# Causal Intervention for Weakly-Supervised Semantic Segmentation

**Dong Zhang**[1]    **Hanwang Zhang**[2]    **Jinhui Tang**[1*]    **Xiansheng Hua**[3]    **Qianru Sun**[4]

[1]School of Computer Science and Engineering, Nanjing University of Science and Technology;
[2]Nanyang Technological University; [3]Damo Academy, Alibaba Group; [4]Singapore Management University.

## Abstract

We present a causal inference framework to improve Weakly-Supervised Semantic Segmentation (WSSS). Specifically, we aim to generate better pixel-level pseudo-masks by using only image-level labels — the most crucial step in WSSS. We attribute the cause of the ambiguous boundaries of pseudo-masks to the confounding context, *e.g.*, the correct image-level classification of "horse" and "person" may be not only due to the recognition of each instance, but also their co-occurrence context, making the model inspection (*e.g.*, CAM) hard to distinguish between the boundaries. Inspired by this, we propose a structural causal model to analyze the causalities among images, contexts, and class labels. Based on it, we develop a new method: Context Adjustment (CONTA), to remove the confounding bias in image-level classification and thus provide better pseudo-masks as ground-truth for the subsequent segmentation model. On PASCAL VOC 2012 and MS-COCO, we show that CONTA boosts various popular WSSS methods to new state-of-the-arts.[1]

## 1   Introduction

Semantic segmentation aims to classify each image pixel into its corresponding semantic class [32]. It is an indispensable computer vision building block for scene understanding applications such as autonomous driving [54] and medical imaging [18]. However, the pixel-level labeling is expensive, *e.g.*, it costs about 1.5 man-hours for one $500 \times 500$ daily life image [12]. Therefore, to scale up, we are interested in *Weakly-Supervised Semantic Segmentation*

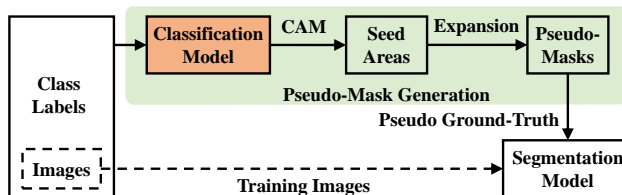

Figure 1: The prevailing pipeline for training WSSS. Our contribution is to improve the Classification Model, which is the foundation for better pseudo-masks.

(WSSS), where the "weak" denotes a much cheaper labeling cost at the instance-level [9, 28] or even at the image-level [23, 57]. In particular, we focus on the latter as it is the most economic way — only a few man-seconds for tagging an image [26].

The prevailing pipeline for training WSSS is depicted in Figure 1. Given training images with only image-level class labels, we first train a multi-label classification model. Second, for each image, we infer the class-specific seed areas, *e.g.*, by applying Classification Activation Map (CAM) [68] to the above trained model. Finally, we expand them to obtain the *Pseudo-Masks* [20, 57, 59], which are

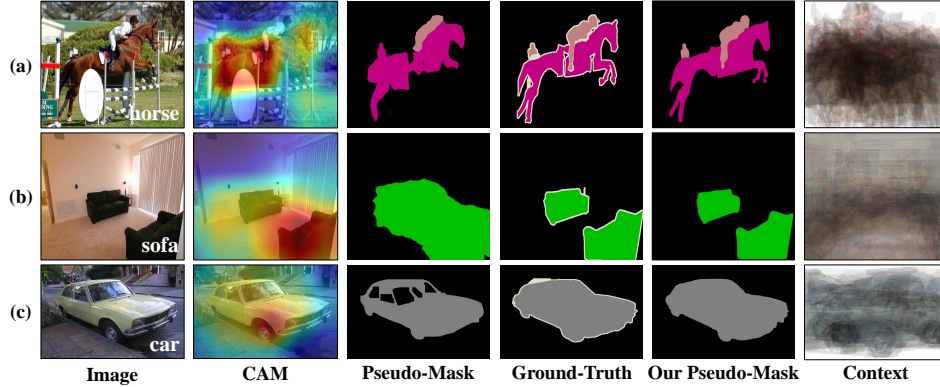

| | Image | CAM | Pseudo-Mask | Ground-Truth | Our Pseudo-Mask | Context |

Figure 2: Three basic problems in existing pseudo-masks [57] (dataset: PASCAL VOC 2012 [12]): (a) Object Ambiguity, (b) Incomplete Background, (c) Incomplete Foreground. They usually combine to cause other complications. The context (mean image per class) may provide clues for the reasons.

used as the pseudo ground-truth for training a standard supervised semantic segmentation model [8]. You might be concerned, there is no free lunch — it is essentially ill-posed to infer pixel-level masks from only image-level labels, especially when the visual scene is complex. Although most previous works have noted this challenge [1, 20, 57], as far as we know, no one answers the whys and wherefores. In this paper, we contribute a formal answer based on causal inference [37] and propose a principled and fundamental solution.

As shown in Figure 2, we begin with illustrating the three basic problems that cause the complications in pseudo-mask generation:

**Object Ambiguity:** Objects are not alone. They usually co-occur with each other under certain contexts. For example, if most "horse" images are about "person riding horse", a classification model will wrongly generalize to "most horses are with people" and hence the generated pseudo-masks are ambiguous about the boundary between "person" and "horse".

**Incomplete Background:** Background is composed of (unlabeled) semantic objects. Therefore, the above ambiguity also holds due to the co-occurrence of foreground and background objects, *e.g.*, some parts of the background "floor" are misclassified as the foreground "sofa".

**Incomplete Foreground:** Some semantic parts of the foreground object, *e.g.*, the "window" of "car", co-vary with different contexts, *e.g.*, the window reflections of the surroundings. Therefore, the classification model resorts to using the less context-dependent (*i.e.*, discriminative) parts to represent the foreground, *e.g.*, the "wheel" part is the most representative of "car".

So far, we can see that all the above problems are due to the context prior in dataset. Essentially, the context is a *confounder* that misleads the image-level classification model to learn spurious correlations between pixels and labels, *e.g.*, the inconsistency between the CAM-expanded pseudo-masks and the ground-truth masks in Figure 2. More specifically, although the confounder is helpful for a better association between the image pixels $X$ and labels $Y$ via a model $P(Y|X)$, *e.g.*, it is likely a "sofa" when seeing a "floor" region, $P(Y|X)$ mistakenly 1) associates non-causal but positively correlated pixels to labels, *e.g.*, the "floor" region wrongly belongs to "sofa", 2) disassociates causal but negatively correlated ones, *e.g.*, the "window" region is wrongly classified as "non-car". To this end, we propose to use $P(Y|do(X))$ instead of $P(Y|X)$ to find what pixels truly cause the labels, where the *do*-operation denotes the pursuit of the causality between the cause $X$ and the effect $Y$ without the confounding effect [39]. The ideal way to calculate $P(Y|do(X))$ is to "physically" intervene $X$ (a.k.a., randomised controlled trial [7]) — if we could have photographed any "sofa" under any context [11], then $P(sofa|do(X)) = P(sofa|X)$. Intrigued, you are encouraged to think about the causal reason why $P(car|X)$ can robustly localize the "wheel" region in Figure 2?[2]

In Section 3.1, we formulate the causalities among pixels, contexts, and labels in a unified Structural Causal Model [36] (see Figure 3 (a)). Thanks to the model, we propose a novel WSSS pipeline called:

Context Adjustment (CONTA). CONTA is based on the backdoor adjustment [37] for $P(Y|do(X))$. Instead of the prohibitively expensive "physical" intervention, CONTA performs a practical "virtual" one from only the observational dataset (the training data *per se*). Specifically, CONTA is an iterative procedure that generates high-quality pseudo-masks. We achieve this by proposing an effective approximation for the backdoor adjustment, which fairly incorporates every possible context into the multi-label classification, generating better CAM seed areas. In Section 4.3, we demonstrate that CONTA can improve pseudo-marks by 2.0% mIoU on average and overall achieves a new state-of-the-art by 66.1% mIoU on the *val* set and 66.7% mIoU on the *test* set of PASCAL VOC 2012 [12], and 33.4% mIoU on the *val* set of MS-COCO [30].

## 2   Related Work

**Weakly-Supervised Semantic Segmentation (WSSS).** To address the problem of expensive labeling cost in fully-supervised semantic segmentation, WSSS has been extensively studied in recent years [1, 59]. As shown in Figure 1, the prevailing WSSS pipeline [23] with only the image-level class labels [2, 57] mainly consists of the following two steps: pseudo-mask generation and segmentation model training. The key is to generate the pseudo-masks as perfect as possible, where the "perfect" means that the pseudo-mask can reveal the entire object areas with accurate boundaries [1]. To this end, existing methods mainly focus on generating better seed areas [25, 57, 59, 58] and expanding these seed areas [1, 2, 20, 23, 55]. In this paper, we also follow this pipeline and our contribution is to propose an iterative procedure to generate high-quality seed areas.

**Visual Context.** Visual context is crucial for recognition [11, 45, 53]. The majority of WSSS models [1, 20, 57, 59] implicitly use context in the backbone network by enlarging the receptive fields with the help of dilated/atrous convolutions [64]. There is a recent work that explicitly uses contexts to improve the multi-label classifier [49]: given a pair of images, it encourages the similarity of the foreground features of the same class and the contrast of the rest. In this paper, we also explicitly use the context, but in a novel framework of causal intervention: the proposed context adjustment.

**Causal Inference.** The purpose of causal inference [39, 43] is to empower models the ability to pursue the causal effect: we can remove the spurious bias [5], disentangle the desired model effects [6], and modularize reusable features that generalize well [35]. Recently, there is a growing number of computer vision tasks that benefit from causality [34, 40, 51, 52, 56, 63, 65]. In our work, we adopt the Pearl's structural causal model [36]. Although the Rubin's potential outcome framework [42] can also be used, as the two are fundamentally equivalent [16, 38], we prefer Pearl's because it can explicitly introduce the causality in WSSS — every node in the graph can be located and implemented in the WSSS pipeline. Nevertheless, we encourage readers to explore Rubin's when some causalities cannot be explicitly hypothesized and modeled, such as using the prospensity scores [3].

## 3   Context Adjustment

Recall in Figure 1 that the pseudo-mask generation is the bottleneck of WSSS, and as we discussed in Section 1, the inaccurate CAM-generated seed areas are due to the context confounder $C$ that misleads the classification model between image $X$ and label $Y$. In this section, we will use a causal graph to fundamentally reveal how the confounder $C$ hurts the pseudo-mask quality (Section 3.1) and how to remove it by using causal intervention (Section 3.2).

### 3.1   Structural Causal Model

We formulate the causalities among pixel-level image $X$, context prior $C$, and image-level labels $Y$, with a Structural Causal Model (SCM) [36]. As illustrated in Figure 3 (a), the direct links denote the causalities between the two nodes: cause $\rightarrow$ effect. Note that the newly added nodes and links other than $X \rightarrow Y^3$ are not deliberately imposed on the original image-level classification; in contrast, they are the ever-overlooked causalities. Now we detail the high-level rationale behind the SCM and defer its implementation in Section 3.2.

$C \rightarrow X$. Context prior $C$ determines what to picture in image $X$. By "context prior", we adopt the general meaning in vision: the relationships among objects in a visual scene [33]. Therefore, $C$ tells us where to put "car", "road", and "building" in an image. Although building a generative model for $C \rightarrow X$ is extremely challenging for complex scenes [22], fortunately, as we will introduce later in Section 3.2, we can avoid it in causal intervention.

$C \rightarrow M \leftarrow X$. $M$ is an image-specific representation using the contextual templates from $C$. For example, a car image can be delineated by using a "car" context template filled with detailed attributes, where the template is the prototypical shape and location of "car" (foreground) in a scene (back-

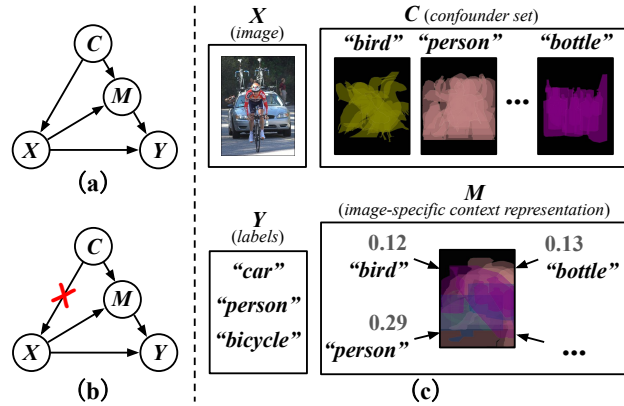

Figure 3: (a) The proposed Structural Causal Model (SCM) for causality of multi-label classifier in WSSS, (b) The intervened SCM for the causality of multi-label classifier in WSSS, (c) The realization of each component in CONTA.

ground). Note that this assumption is not *ad hoc* in our model, in fact, it underpins almost every concept learning method from the classic Deformable Part Models [13] to modern CNNs [15], whose cognitive evidence can be found in [24]. A plausible realization of $M$ and $C$ used in Section 3.2 is illustrated in Figure 3 (c).

$X \rightarrow Y \leftarrow M$. A general $C$ cannot directly affect the labels $Y$ of an image. Therefore, besides the conventional classification model $X \rightarrow Y$, $Y$ is also the effect of the $X$-specific mediation $M$. $M \rightarrow Y$ denotes an obvious causality: the contextual constitution of an image affects the image labels. It is worth noting that even if we do not explicitly take $M$ as an input for the classification model, $M \rightarrow Y$ still holds. The evidence lies in the fact that visual contexts will emerge in higher-level layers of CNN when training image classifiers [66, 68], which essentially serve as a feature map backbone for modern visual detection that highly relies on contexts, such as Fast R-CNN [14] and SSD [31]. To think conversely, if $M \nrightarrow Y$ in Figure 3 (a), the only path left from $C$ to $Y$: $C \rightarrow X \rightarrow Y$, is cut off conditional on $X$, then no contexts are allowed to contribute to the labels by training $P(Y|X)$, and thus we would never uncover the context, *e.g.*, the seed areas. So, WSSS would be impossible.

So far, we have pinpointed the role of context $C$ played in the causal graph of image-level classification in Figure 3 (a). Thanks to the graph, we can clearly see how $C$ confounds $X$ and $Y$ via the backdoor path $X \leftarrow C \rightarrow M \rightarrow Y$: even if some pixels in $X$ have nothing to do with $Y$, the backdoor path can still help to correlate $X$ and $Y$, resulting the problematic pseudo-masks in Figure 2. Next, we propose a causal intervention method to remove the confounding effect.

## 3.2 Causal Intervention via Backdoor Adjustment

We propose to use causal intervention: $P(Y|do(X))$, as the new image-level classifier, which removes the confounder $C$ and pursues the true causality from $X$ to $Y$ so as to generate better CAM seed areas. As the "physical" intervention — collecting objects in any context — is impossible, we apply the backdoor adjustment [39] to "virtually" achieve $P(Y|do(X))$. The key idea is to 1) cut off the link $C \rightarrow X$ in Figure 3 (b), and 2) stratify $C$ into pieces $C = \{c\}$. Formally, we have:

$$P(Y|do(X)) = \sum_c P(Y|X, M = f(X, c)) P(c), \tag{1}$$

where $f(\cdot)$ is a function defined later in Eq. (3). As $C$ is no longer correlated with $X$, the causal intervention makes $X$ have a fair opportunity to incorporate every context $c$ into $Y$'s prediction, subject to a prior $P(c)$.

However, $C$ is not observable in WSSS, let alone stratifying it. To this end, as illustrated in Figure 3 (c), we use the class-specific average mask in our proposed Context Adjustment (CONTA) to approximate the confounder set $C = \{c_1, c_2, ..., c_n\}$, where $n$ is the class size in dataset and $c \in \mathbb{R}^{h \times w}$ corresponds to the $h \times w$ average mask of the $i$-th class images. $M$ is the $X$-specific mask which can be viewed

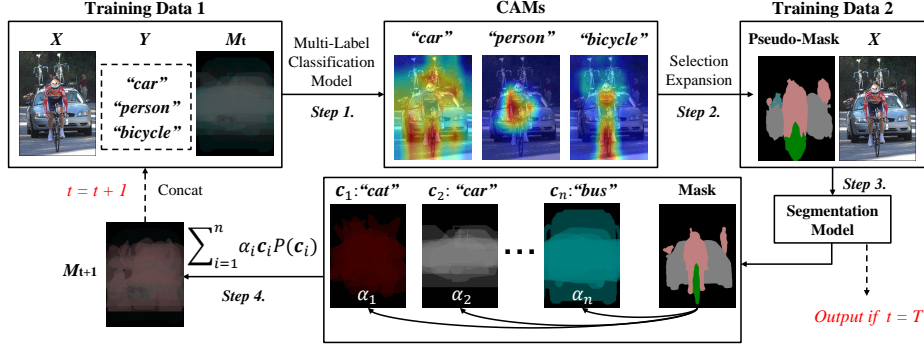

Figure 4: Overview of our proposed Context Adjustment (CONTA). $M_t$ is an empty set when $t = 0$.

as a linear combination of $\{c\}$. Note that the rationale behind our $C$'s implementation is based on the definition of context: the relationships among the objects [33], and thus each stratification is about one class of object interacting with others (*i.e.*, the background). So far, how do we obtain the unobserved masks? In CONTA, we propose an iterative procedure to establish the unobserved $C$.

Figure 4 illustrates the overview of CONTA. The input is training images with only class labels (**Training Data 1**, $t = 0$), the output is a segmentation model ($t = T$), which is trained on CONTA generated pseudo-masks (**Training Data 2**). Before we delve into the steps below, we highlight that CONTA is essentially an EM algorithm [60], if you view Eq. (1) as an objective function (where we omit the model parameter $\Theta$) of observed data $X$ and missing data $C$. Thus, its convergence is theoretically guaranteed. As you may realize soon, the E-step is to calculate the expectation ($\sum_c$ in Eq. (1)) over the estimated masks in $C|(X, \Theta_t)$ (**Step 2, 3, 4**); and the M-step is to maximize Eq. (1) for $\Theta_{t+1}$ (**Step 1**).

***Step 1***. **Image Classification.** We aim to maximize $P(Y|do(X))$ for learning the multi-label classification model, whereby the subsequent CAM will yield better seed areas. Our implementation for Eq. (1) is:

$$P(Y|do(X); \Theta_t) = \prod_{i=1}^{n} \left[ \mathbb{1}_{i \in Y} \frac{1}{1 + \exp(-s_i)} + \mathbb{1}_{i \notin Y} \frac{1}{1 + \exp(s_i)} \right], \tag{2}$$

where $\mathbb{1}$ is 1/0 indicator, $s_i = f(X, M_t; \theta_t^i)$ is the $i$-th class score function, consisting of a class-shared convolutional network on the channel-wise concatenated feature maps $[X, M_t]$, followed by a class-specific fully-connected network (the last layer is based on a global average pooling [29]). Overall, Eq. (2) is a joint probability over all the $n$ classes that encourages the ground-truth labels $i \in Y$ and penalizes the opposite $i \notin Y$. In fact, the negative log-likelihood loss of Eq. (2) is also known as the multi-label soft-margin loss [44]. Note that the expectation $\sum_c$ is absorbed in $M_t$, which will be detailed in ***Step 4***.

***Step 2***. **Pseudo-Mask Generation.** For each image, we can calculate a set of class-specific CAMs [68] using the trained classifier above. Then, we follow the conventional two post-processing steps: 1) We select hot CAM areas (subject to a threshold) for seed areas [2, 57]; and 2) We expand them to be the final pseudo-masks [1, 23].

***Step 3***. **Segmentation Model Training.** Each pseudo-mask is used as the pseudo ground-truth for training any standard supervised semantic segmentation model. If $t = T$, this is the model for delivery; otherwise, its segmentation mask can be considered as an additional post-processing step for pseudo-mask smoothing. For fair comparisons with other WSSS methods, we adopt the classic DeepLab-v2 [8] as the supervised semantic segmentation model. Performance boost is expected if you adopt more advanced ones [27].

***Step 4***. **Computing $M_{t+1}$.** We first collect the predicted segmentation mask $X_m$ of every training image from the above trained segmentation model. Then, each class-specific entry $c$ in the confounder set $C$ is the averaged mask of $X_m$ within the corresponding class and is reshaped into a $hw \times 1$ vector. So far, we are ready to calculate Eq. (1). However, the cost of the network forward pass for all the $n$ classes is expensive. Fortunately, under practical assumptions (see Appendix 2), we can adopt the Normalized Weighted Geometric Mean [62, 63] to move the outer sum $\sum_c P(\cdot)$ into the feature level: $\sum_c P(Y|X, M)P(c) \approx P(Y|X, M = \sum_c f(X, c)P(c))$, thus, we only need to feed-forward

the network once. We have:

$$M_{t+1} = \sum_{i=1}^{n} \alpha_i c_i P(c_i), \ \alpha_i = softmax\left(\frac{(\mathbf{W}_1 X_m)^T (\mathbf{W}_2 c_i)}{\sqrt{n}}\right), \quad (3)$$

where $\alpha_i$ is the normalized similarity (softmax over $n$ similarities) between $X_m$ and the $i$-th entry $c_i$ in the confounder set $C$. To make CONTA beyond the dataset statistics *per se*, $P(c_i)$ is set as the uniform $1/n$. $\mathbf{W}_1, \mathbf{W}_2 \in \mathbb{R}^{n \times hw}$ are two learnable projection matrices, which are used to project $X_m$ and $c_i$ into a joint space. $\sqrt{n}$ is a constant scaling factor that is used as for feature normalization as in [56].

## 4 Experiments

We evaluated the proposed CONTA in terms of the model performance quantitatively and qualitatively. Below we introduce the datasets, evaluation metric, and baseline models. We demonstrate the ablation study, show the effectiveness of CONTA on different baselines, and compare it to the state-of-the-arts. Further details and results are given in Appendix.

### 4.1 Settings

**Datasets.** PASCAL VOC 2012 [12] contains 21 classes (one background class) which includes 1,464, 1,449 and 1,456 images for *training*, *validation* (*val*) and *test*, respectively. As the common practice in [1, 57], in our experiments, we used an enlarged training set with 10,582 images, where the extra images and labels are from [17]. MS-COCO [30] contains 81 classes (one background class), 80k, and 40k images for *training* and *val*. Although pixel-level labels are provided in these benchmarks, we only used image-level class labels in the training process.

**Evaluation Metric.** We evaluated three types of masks: CAM seed area mask, pseudo-mask, and segmentation mask, compared with the ground-truth mask. The standard mean Intersection over Union (mIoU) was used on the *training* set for evaluating CAM seed area mask and pseudo-mask, and on the *val* and *test* sets for evaluating segmentation mask.

**Baseline Models.** To demonstrate the applicability of CONTA, we deployed it on four popular WSSS models including one seed area generation model: SEAM [57], and three seed area expansion models: IRNet [1], DSRG [20], and SEC [23]. Specially, DSRG requires the extra saliency mask [21] as the supervision. General architecture components include a multi-label image classification model, a pseudo-mask generation model, and a segmentation model: DeepLab-v2 [8]. Since the experimental settings of them are different, for fair comparison, we adopted the same settings as reported in the official codes. The detailed implementations of each baseline + CONTA are given in Appendix 3.

### 4.2 Ablation Study

Our ablation studies aim to answer the following questions. **Q1**: *Does CONTA merely take the advantage of the mask refinement? Is $M_t$ indispensable?* We validated these by concatenating the segmentation mask (which is more refined compared to the pseudo-mask) with the backbone feature map, fed into classifiers. Then, we compared the newly generated results with the baseline ones. **Q2**: *How many rounds?* We recorded the performances of CONTA in each round. **Q3**: *Where to concatenate $M_t$?* We adopted the channel-wise feature map concatenation $[X, M_t]$ on different blocks of the backbone feature maps and tested which block has the most

| | Setting | CAM | Pseudo-Mask | Seg. Mask |
|---|---|---|---|---|
| | Upperbound [32] | – | – | 80.8 |
| | Baseline* [57] | 55.1 | 63.1 | 64.3 |
| (Q1) | $M_t \leftarrow$ Seg. Mask | 55.0 | 62.7 | 64.0 |
| (Q2) | Round = 1 | 55.6 | 64.2 | 65.0 |
| | Round = 2 | 55.9 | 64.8 | 65.8 |
| | Round = 3 | **56.2** | **65.4** | **66.1** |
| | Round = 4 | 56.1 | 64.8 | 65.5 |
| (Q3) | Block-2 | 55.5 | 64.3 | 65.2 |
| | Block-3 | 55.6 | 64.5 | 65.3 |
| | Block-4 | 56.0 | 65.1 | 65.9 |
| | Block-5 | **56.2** | **65.4** | **66.1** |
| | Dense | 56.1 | **65.4** | 66.0 |
| (Q4) | $C_{\text{Pseudo-Mask}}$ | 56.0 | 65.2 | 65.8 |
| | $C_{\text{Seg. Mask}}$ | **56.2** | **65.4** | **66.1** |

Table 1: Ablation results on PASCAL VOC 2012 [12] in mIoU (%). "*" denotes our re-implemented results. "Seg. Mask" refers to the segmentation mask on the *val* set. "–" denotes that it is N.A. for the fully-supervised models.

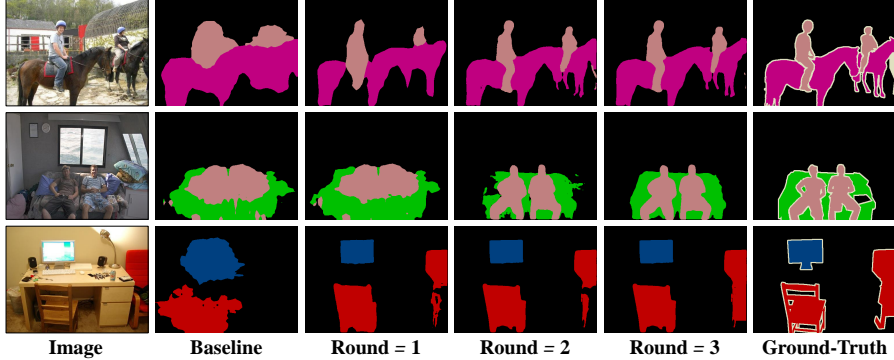

| Image | Baseline | Round = 1 | Round = 2 | Round = 3 | Ground-Truth |

Figure 5: Visualization of pseudo-masks (baseline: SEAM [57], dataset: PASCAL VOC 2012 [12]).

improvement. **Q4**: *What is in the confounder set?* We compared the effectiveness of using the pseudo-mask and the segmentation mask to construct the confounder set $C$.

Due to page limit, we only showed ablation studies on the state-of-the-art WSSS model: SEAM [57], and the commonly used dataset – PASCAL VOC 2012; other methods on MS-COCO are given in Appendix 4. We treated the performance of the fully-supervised DeepLab-v2 [8] as the upperbound.

**A1**: Results in Table 1 (**Q1**) show that using the segmentation mask instead of the proposed $M_t$ (concatenated to block-5) is even worse than the baseline. Therefore, the superiority of CONTA is not merely from better (smoothed) segmentation masks and $M_t$ is empirically indispensable.

**A2**: Here, $[X, M_t]$ was applied to block-5, and the segmentation masks were used to establish the confounder set $C$. From Table 1 (**Q2**), we can observe that the performance starts to saturated at round 3. In particular, when round = 3, CONTA can achieve the unanimously best mIoU on CAM, pseudo-mask, and segmentation mask. Therefore, we set #round = 3 in the following CONTA experiments. We also visualized some qualitative results of the pseudo-masks in Figure 5. We can observe that CONTA can gradually segment clearer boundaries when compared to the baseline results, *e.g.*, person's leg vs. horse, person's body vs. sofa, chair's leg vs. background, and horse's leg vs. background.

**A3**: In addition to $[X, M_t]$ on various backbone blocks, we also reported a dense result, *i.e.*, $[X, M_t]$ on block-2 to block-5. In particular, $[X, M_t]$ was concatenated to the last layer of each block. Before the feature map concatenation, the map size of $M_t$ should be down-sampled to match the corresponding block. Results in Table 1 (**Q3**) show that the performance at block-2/-3 are similar, and block-4/-5 are slightly higher. In particular, when compared to the baseline, block-5 has the most mIoU gain by 1.1% on CAM, 2.3% on pseudo-mask, and 1.8% on segmentation mask. One possible reason is that feature maps at block-5 contain higher-level contexts (*e.g.*, bigger parts, and more complete boundaries), which are more consistent with $M_t$, which are essential contexts. Therefore, we applied $[X, M_t]$ on block-5.

**A4**: From Table 1 (**Q4**), we can observe that using both of the pseudo-mask and the segmentation mask established $C$ ($C_{\text{Pseudo-Mask}}$ and $C_{\text{Seg. Mask}}$) can boost the performance when compared to the baseline. In particular, the segmentation mask has a larger gain. The reason may be that the trained segmentation model can smooth the pseudo-mask and thus using higher-quality masks to approximate the unobserved confounder set is better.

| Method | Backbone | CAM | Pseudo-Mask | Seg. Mask |
|---|---|---|---|---|
| SEC [23] | VGG-16 | 46.5 | 53.4 | 50.7 |
| + CONTA | VGG-16 | $47.9_{+1.4}$ | $55.7_{+2.3}$ | $53.2_{+2.5}$ |
| SEAM* [57] | ResNet-38 | 55.1 | 63.1 | 64.3 |
| + CONTA | ResNet-38 | $\mathbf{56.2}_{+1.1}$ | $65.4_{+2.3}$ | $\mathbf{66.1}_{+1.8}$ |
| IRNet* [1] | ResNet-50 | 48.3 | 65.9 | 63.0 |
| + CONTA | ResNet-50 | $48.8_{+0.5}$ | $\mathbf{67.9}_{+2.0}$ | $65.3_{+2.3}$ |
| DSRG [20] | ResNet-101 | 47.3 | 62.7 | 61.4 |
| + CONTA | ResNet-101 | $48.0_{+0.7}$ | $64.0_{+1.3}$ | $62.8_{+1.4}$ |

Table 2: Different baselines+CONTA on PASCAL VOC 2012 [12] dataset in mIoU (%). "*" denotes our re-implemented results. "Seg. Mask" refers to the segmentation mask on the *val* set.

## 4.3 Effectiveness on Different Baselines

To demonstrate the applicability of CONTA, in addition to SEAM [57], we also deployed CONTA on IRNet [1], DSRG [20], and SEC [23]. In particular, the round was set to 3 for SEAM, IRNet and

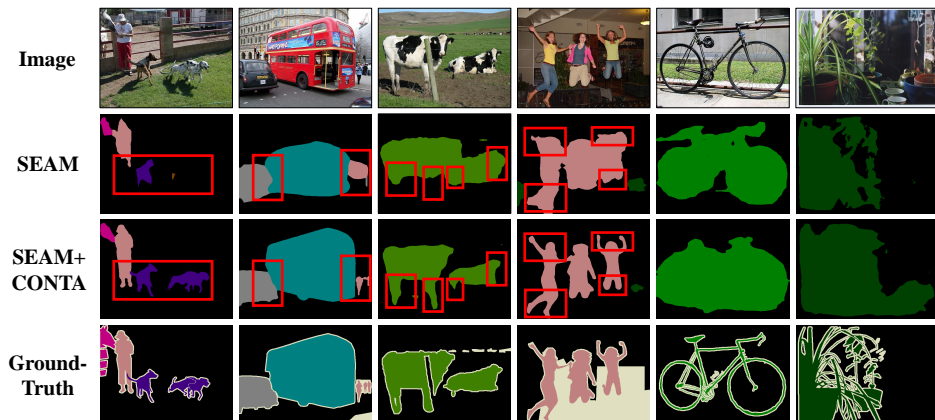

Figure 6: Visualization of segmentation masks, the last two columns show two failure cases (dataset: PASCAL VOC 2012 [12]). The red rectangle highlights the better areas for SEAM+CONTA.

SEC, and was set to 2 for DSRG. Experimental results on PASCAL VOC 2012 are shown in Table 2. We can observe that deploying CONTA on different WSSS models improve all their performances. There are the averaged mIoU improvements of 0.9% on CAM, 2.0% on pseudo-mask, and 2.0% on segmentation mask. In particular, CONTA deployed on SEAM can achieve the best performance of 56.2% on CAM and 66.1% on segmentation mask. Besides, CONTA deployed on IRNet can achieve the best performance of 67.9% on the pseudo-mask. The above results demonstrate the applicability and effectiveness of CONTA.

## 4.4 Comparison with State-of-the-arts

Table 3 lists the overall WSSS performances. On PASCAL VOC 2012, we can observe that CONTA deployed on IRNet with ResNet-50 [19] achieves the very competitive 65.3% and 66.1% mIoU on the *val* set and the *test* set. Based on a stronger backbone ResNet-38 [61] (with fewer layers but wider

| Method | Backbone | *val* | *test* |
|---|---|---|---|
| AffinityNet [2] | ResNet-38 | 61.7 | 63.7 |
| RRM [67] | ResNet-38 | 62.6 | 62.9 |
| SSDD [47] | ResNet-38 | **64.9** | 65.5 |
| SEAM [57] | ResNet-38 | 64.5 | **65.7** |
| IRNet [1] | ResNet-50 | 63.5 | 64.8 |
| IRNet+CONTA | ResNet-50 | 65.3 | 66.1 |
| SEAM+CONTA | ResNet-38 | **66.1** | **66.7** |

(a) PASCAL VOC 2012 [12].

| Method | Backbone | *val* |
|---|---|---|
| BFBP [45] | VGG-16 | 20.4 |
| SEC [23] | VGG-16 | 22.4 |
| SEAM* [57] | ResNet-38 | 31.9 |
| IRNet* [1] | ResNet-50 | **32.6** |
| SEC+CONTA | VGG-16 | 23.7 |
| SEAM+CONTA | ResNet-38 | 32.8 |
| IRNet+CONTA | ResNet-50 | **33.4** |

(b) MS-COCO [30].

Table 3: Comparison with state-of-the-arts in mIoU (%). "*" denotes our re-implemented results. The **best** and **second best** performance under each set are marked with corresponding formats.

channels), CONTA on SEAM achieves state-of-the-art 66.1% and 66.7% mIoU on the *val* set and the *test* set, which surpasses the previous best model 1.2% and 1.0%, respectively. On MS-COCO, CONTA deployed on SEC with VGG-16 [48] achieves 23.7% mIoU on the *val* set, which surpasses the previous best model by 1.3% mIoU. Besides, on stronger backbones and WSSS models, CONTA can also boost the performance by 0.9% mIoU on average.

Figure 6 shows the qualitative segmentation mask comparisons between SEAM+CONTA and SEAM [57]. From the first four columns, we can observe that CONTA can make more accurate predictions on object location and boundary, *e.g.*, person's leg, dog, car, and cow's leg. Besides, we also show two failure cases of SEAM+CONTA in the last two columns, where bicycle and plant can not be well predicted. One possible explanation is that the segmentation mask is directly obtained from the 8× down-sampled feature maps, so some complex-contour objects can not be accurately delineated. This problem may be alleviated by using the encoder-decoder segmentation model, *e.g.*, SegNet [4], and U-Net [41]. More visualization results are given in Appendix 5.

# 5  Conclusion

We started from summarizing the three basic problems in existing pseudo-masks of WSSS. Then, we argued that the reasons are due to the context prior, which is a confounder in our proposed causal graph. Based on the graph, we used causal intervention to remove the confounder. As it is unobserved, we devised a novel WSSS framework: Context Adjustment (CONTA), based on the backdoor adjustment. CONTA can promote all the prevailing WSSS methods to the new state-of-the-arts. Thanks to the causal inference framework, we clearly know the limitations of CONTA: the approximation of the context confounder, which is proven to be ill-posed [10]. Therefore, as moving forward, we are going to 1) develop more advanced confounder set discovery methods and 2) incorporate observable expert knowledge into the confounder.

## Acknowledgements

The authors would like to thank all the anonymous reviewers for their constructive comments and suggestions. This work was partially supported by the National Key Research and Development Program of China under Grant 2018AAA0102002, the National Natural Science Foundation of China under Grant 61732007, the China Scholarships Council under Grant 201806840058, the Alibaba Innovative Research (AIR) programme, and the NTU-Alibaba JRI.

## Broader Impact

The positive impacts of this work are two-fold: 1) it improves the fairness of the weakly-supervised semantic segmentation model, which can prevent the potential discrimination of deep models, *e.g.*, an unfair AI could blindly cater to the majority, causing gender, racial or religious discrimination; 2) it allows some objects to be accurately segmented without extensive multi-context training images, *e.g.*, to segment a car on the road, by using our proposed method, we don't need to photograph any car under any context. The negative impacts could also happen when the proposed weakly-supervised semantic segmentation technique falls into the wrong hands, *e.g.*, it can be used to segment the minority groups for malicious purposes. Therefore, we have to make sure that the weakly-supervised semantic segmentation technique is used for the right purpose.

## Footnotes

[1]Code is open-sourced at: https://github.com/ZHANGDONG-NJUST/CONTA

[2]**Answer:** the "wheel" was photographed in every "car" under any context by the dataset creator.

[3]Some studies [46] show that label causes image ($X \leftarrow Y$). We believe that such anti-causal assumption only holds when the label is as simple as the disentangled causal mechanisms [35, 50] (*e.g.*, 10-digit in MNIST).

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
