[Supplementary Material]

# Appendix for "Causal Intervention for Weakly-Supervised Semantic Segmentation"

This appendix includes the derivation of backdoor adjustment for the proposed structural causal model (Section 1), the normalized weighted geometric mean (Section 2), the detailed implementations for different baseline models (Section 3), the supplementary ablation studies (Section 4), and more visualization results of segmentation masks (Section 5).

## 1 Derivation of Backdoor Adjustment for the Proposed Causal Graph

In the main paper, we used backdoor adjustment [16] to perform the causal intervention. In this section, we show the derivation of backdoor adjustment for the proposed causal graph (in Figure 3(b) of the main paper), by leveraging the following three $do$-calculus rules [15].

Given an arbitrary causal directed acyclic graph $\mathcal{G}$, there are four nodes respectively represented by $X, Y, Z$, and $W$. Particularly, $\mathcal{G}_{\overline{X}}$ denotes the intervened causal graph where all *incoming* arrows to $X$ are deleted, and $\mathcal{G}_{\underline{X}}$ denotes another intervened causal graph where all *outgoing* arrows from $X$ are deleted. We use the lower cases $x$, $y$, $z$, and $w$ to represent the respective values of nodes: $X = x, Y = y, Z = z$, and $W = w$. For any interventional distribution compatible with $\mathcal{G}$, we have the following three rules:

**Rule 1.** Insertion/deletion of observations:
$$P(y|do(x), z, w) = P(y|do(x), w), \text{if } (YZ|X, W)_{\mathcal{G}_{\overline{X}}}. \tag{A1}$$

**Rule 2.** Action/observation exchange:
$$P(y|do(x), do(z), w) = P(y|do(x), z, w), \text{if } (YZ|X, W)_{\mathcal{G}_{\overline{X}\underline{Z}}}. \tag{A2}$$

**Rule 3.** Insertion/deletion of actions:
$$P(y|do(x), do(z), w) = P(y|do(x), w), \text{if } (YZ|X, W)_{\mathcal{G}_{\overline{X Z(W)}}}, \tag{A3}$$

where $Z(W)$ is a subset of $Z$ that are not ancestors of any specific nodes related to $W$ in $\mathcal{G}_{\underline{X}}$. Based on these three rules, we can derive the interventional distribution $P(Y|do(X))$ for our proposed causal graph (in Figure 3(b) of the main paper) by:

$$P(Y|do(X)) = \sum_c P(Y|do(X), c)P(c|do(X)) \tag{A4}$$

$$= \sum_c P(Y|do(X), c)P(c) \tag{A5}$$

$$= \sum_c P(Y|X, c)P(c) \tag{A6}$$

$$= \sum_c P(Y|X, c, M)P(M|X, c)P(c) \tag{A7}$$

$$= \sum_c P(Y|X, c, M = f(X, c))P(c) \tag{A8}$$

$$= \sum_c P(Y|X, M = f(X, c))P(c), \tag{A9}$$

where Eq. A4 and Eq. A7 follow the law of total probability. We can obtain Eq. A5 via **Rule 3** that given $cX$ in $\mathcal{G}_{\overline{X}}$, and Eq. A6 can be obtained via **Rule 2** which changes the intervention term into observation as $YX|c$ in $\mathcal{G}_{\underline{X}}$. Eq. A8 is because in our causal graph, $M$ is an image-specific context representation given by the function $f(X, c)$, and Eq. A9 is essentially equal to Eq. A8.

## 2 Normalized Weighted Geometric Mean

This is Appendix to Section 3.2 "Step 4. Computing $M_{t+1}$". In Section 3.2 of the main paper, we used the Normalized Weighted Geometric Mean (NWGM) [21] to move the outer sum $\sum_c P(\cdot)$ into the feature level: $\sum_c P(Y|X, M)P(c) \approx P(Y|X, M = \sum_c f(X, c)P(c))$. Here, we show the detailed derivation. Formally, our implementation for the positive term (*i.e.*, $1_{i \in Y}$ in Eq.(2) of the main paper) can be derived by:

$$P(Y|do(X)) = \sum_c \frac{exp(s_1(c))}{exp(s_1(c)) + exp(s_2(c))} P(c) \tag{A10}$$

$$= \sum_c Softmax(s_1(c))P(c) \tag{A11}$$

$$\approx \text{NWGM}(Softmax(s_1(c))) \tag{A12}$$

$$= \frac{\prod_c [exp(s_1(c)]^{P(c)}}{\prod_c [exp(s_1(c)]^{P(c)} + \prod_c [exp(s_2(c)]^{P(c)}} \tag{A13}$$

$$= \frac{exp(\sum_c(s_1(c)P(c)))}{exp(\sum_c(s_1(c)P(c))) + exp(\sum_c(s_2(c)P(c)))} \tag{A14}$$

$$= \frac{exp(\mathbb{E}_c(s_1(c)))}{exp(\mathbb{E}_c(s_1(c))) + exp(\mathbb{E}_c(s_2(c)))} \tag{A15}$$

$$= Softmax(\mathbb{E}_c(s_1(c)), \tag{A16}$$

where $s_1(\cdot)$ denotes the positive predicted score for the class label which is indeed associated with the input image, and $s_2(c) = 0$ under this condition. We can obtain Eq. A10 via our implementation of the multi-label image classification model, and obtain Eq. A11 and Eq. A16 via the definition of the softmax function. Eq. A12 can be obtained via the results in [3]. Eq. A13 to Eq. A15 follow the derivation in [21]. Since $s_1(\cdot)$ in our implementation is a linear model, we can use Eq.(3) in the main paper to compute $M_{t+1}$. In addition to the positive term, we can also obtain derivation for the negative term (*i.e.*, $1_{i \notin Y}$ in Eq.(2) of the main paper) through the similar process as above.

## 3 More Implementation Details

This is Appendix to Section 4.1 "Settings". In Section 4.1 of the main paper, we deployed CONTA on four popular WSSS models including SEAM [19], IRNet [1], DSRG [7], and SEC [10]. In this section, we show the detailed implementations of these four models.

### 3.1 Implementation of SEAM+CONTA

**Backbone.** ResNet-38 [20] was adopted as the backbone network. It was pre-trained on ImageNet [4] and its convolution layers of the last three blocks were replaced by dilated convolutions [22] with a common input stride of 1 and their dilation rates were adjusted, such that the backbone network can return a feature map of stride 8, *i.e.*, the output size of the backbone network was $1/8$ of the input.

**Setting.** The input images were randomly re-scaled in the range of $[448, 768]$ by the longest edge and then cropped into a fix size of $448 \times 448$ using zero padding if needed.

**Training Details.** The initial learning rate was set to $0.01$, following the poly policy $lr_{init} = lr_{init}(1 - itr/max\_itr)^\rho$ with $\rho = 0.9$ for decay. Online hard example mining [17] was employed on the training loss to preserve only the top $20\%$ pixel losses. The model was trained with batch size as 8 for 8 epochs using Adam optimizer [9]. We deployed the same data augmentation strategy (*i.e.*, horizontal flip, random cropping, and color jittering [12]), as in AffinityNet [2], in our training process.

**Hyper-parameters.** The hard threshold parameter for CAM was set to 16 by default and changed to 4 and 24 to amplify and weaken background activation, respectively. The fully-connected CRF [11] was used to refine CAM, pseudo-mask, and segmentation mask with the default parameters in the

public code. For seed areas expansion, the AffinityNet [2] was used with the search radius as $\gamma = 5$, the hyper-parameter in the Hadamard power of the affinity matrix as $\beta = 8$, and the number of iterations in random walk as $t = 256$.

## 3.2 Implementation of IRNet+CONTA

**Backbone.** ResNet-50 [6] was used as the backbone network (pre-trained on ImageNet [4]). The adjusted dilated convolutions [22] were used in the last two blocks with a common input stride of 1, such that the backbone network can return a feature map of stride 16, *i.e.*, the output size of the backbone network was $1/16$ of the input.

**Setting.** The input image was cropped into a fix size of $512 \times 512$ using zero padding if needed.

**Training Details.** The stochastic gradient descent was used for optimization with $8,000$ iterations. Learning rate was initially set to $0.1$, and decreased using polynomial decay $lr_{init} = lr_{init}(1 - itr/max\_itr)^\rho$ with $\rho = 0.9$ at every iteration. The batch size was set to 16 for the image classification model and 32 for the inter-pixel relation model. The same data augmentation strategy (*i.e.*, horizontal flip, random cropping, and color jittering [12]) as in AffinityNet [2] was used in the training process.

**Hyper-parameters.** The fully-connected CRF [11] was used to refine CAM, pseudo-mask, and segmentation mask with the default parameters given in the original code. The hard threshold parameter for CAM was set to 16 by default and changed to 4 and 24 to amplify and weaken the background activation, respectively. The radius $\gamma$ that limits the search space of pairs was set to 10 when training, and reduced to 5 at inference (conservative propagation in inference). The number of random walk iterations $t$ was fixed to 256. The hyper-parameter $\beta$ in the Hadamard power of the affinity matrix was set to 10.

## 3.3 Implementation of DSRG+CONTA

**Backbone.** ResNet-101 [6] was used as the backbone network (pre-trained on ImageNet [4]) where dilated convolutions [22] were used in the last two blocks, such that the backbone network can return a feature map of stride 16, *i.e.*, the output size of the backbone network was $1/16$ of the input.

**Setting.** The input image was cropped into a fix size of $321 \times 321$ using zero padding if needed.

**Training Details.** The stochastic gradient descent with mini-batch was used for network optimization with $10,000$ iterations. The momentum and the weight decay were set to $0.9$ and $0.0005$, respectively. The batch size was set to 20, and the dropout rate was set to $0.5$. The initial learning rate was set to $0.0005$ and it was decreased by a factor of 10 every $2,000$ iterations.

**Hyper-parameters.** For seed generation, pixels with the top $20\%$ activation values in the CAM were considered as foreground (objects) as in [23]. For saliency masks, the model in [8] was used to produce the background localization cues with the normalized saliency value $0.06$. For the similarity criteria, the foreground threshold and the background threshold were set to $0.99$ and $0.85$, respectively. The fully-connected CRF [11] was used to refine pseudo-mask and segmentation mask with the default parameters in the public code.

## 3.4 Implementation of SEC+CONTA

**Backbone.** VGG-16 [18] was used as the backbone network (pre-trained on ImageNet [4]), where the last two fully-connected layers were substituted with randomly initialized convolutional layers, which have 1024 output channels and kernels of size 3, such that the output size of the backbone network was $1/8$ of the input.

**Setting.** The input image was cropped into a fix size of $321 \times 321$ using zero padding if needed.

**Training Details.** The weights for the last (prediction) layer were randomly initialized from a normal distribution with mean 0 and variance $0.01$. The stochastic gradient descent was used for the network optimization with $8,000$ iterations, the batch size was set to 15, the dropout rate was set to $0.5$ and the weight decay parameter was set to $0.0005$. The initial learning rate was $0.001$ and it was decreased by a factor of 10 every $2,000$ iterations.

| | Setting | CAM | Pseudo-Mask | Seg. Mask |
|---|---|---|---|---|
| | Upperbound [14] | – | – | 72.3 |
| | Baseline* [1] | 48.3 | 65.9 | 63.0 |
| **(A1)** | $M_t \leftarrow$ Seg. Mask | 48.1 | 65.5 | 62.1 |
| | Round = 1 | 48.5 | 66.9 | 64.2 |
| **(A2)** | Round = 2 | 48.7 | 67.6 | 65.0 |
| | Round = 3 | **48.8** | **67.9** | **65.3** |
| | Round = 4 | 48.6 | 67.2 | 64.9 |
| | Block-2 | 48.3 | 66.2 | 63.4 |
| | Block-3 | 48.4 | 66.6 | 63.8 |
| **(A3)** | Block-4 | 48.7 | 67.3 | 64.6 |
| | Block-5 | **48.8** | **67.9** | **65.3** |
| | Dense | 48.7 | 67.6 | 65.1 |
| **(A4)** | $C_{\text{Pseudo-Mask}}$ | 48.6 | 67.4 | 65.0 |
| | $C_{\text{Seg. Mask}}$ | **48.8** | **67.9** | **65.3** |

Table A1: Ablations of IRNet [1]+CONTA on PASCAL VOC 2012 [5] in mIoU (%). "*" denotes our re-implemented results. "Seg. Mask" refers to the segmentation mask of the *val* set. "–" denotes that the result is N.A. for the fully-supervised model.

**Hyper-parameters.** For seed generation, pixels with the top $20\%$ activation values in the CAM were considered as foreground (objects) as in [23]. The fully-connected CRF [11] was used to refine pseudo-mask and segmentation mask with the spatial distance was multiplied by 12 to reflect the fact that the original image was down-scaled to match the size of the predicted segmentation mask, and the other parameters are consistent with the public code.

# 4 More Ablation Study Results

This is Appendix to Section 4.2 "Ablation Study". In Section 4.2 of the main paper, we showed the ablation study results of SEAM [19]+CONTA on PASCAL VOC 2012 [5]. In this section, we show the results of IRNet [1]+CONTA, DSRG [7]+CONTA, and SEC [10]+CONTA on PASCAL VOC 2012. Besides, we also show the results of SEAM+CONTA, IRNet+CONTA, DSRG+CONTA, and SEC+CONTA on MS-COCO [13].

## 4.1 PASCAL VOC 2012

Table A1, Table A2, and Table A3 show ablation results of IRNet+CONTA, DSRG+CONTA, and SEC+CONTA on PASCAL VOC 2012, respectively. We can observe that IRNet+CONTA and SEC+CONTA can achieve the best performance at round= 3, and DSRG+CONTA can achieve the best mIoU score at round= 2. In addition to results of SEAM+CONTA in our main paper, we can see that IRNet+CONTA can achieve the second best mIoU results: $48.8\%$ on CAM, $67.9\%$ on pseudo-mask, and $65.3\%$ on segmentation mask.

## 4.2 MS-COCO

Table A4, Table A5, Table A6, and Table A7 show the respective ablation results of SEAM+CONTA, IRNet+CONTA, DSRG+CONTA, and SEC+CONTA on MS-COCO. We can see that SEAM+CONTA, IRNet+CONTA and, SEC+CONTA can achieve the top mIoU at round= 3, and DSRG+CONTA can achieve the best performance at round= 2. In particular, we see that the mIoU scores of IRNet+CONTA are the best on MS-COCO as respectively $28.7\%$ on CAM, $35.2\%$ on pseudo-mask, and $33.4\%$ on segmentation mask.

| | Setting | CAM | Pseudo-Mask | Seg. Mask |
|---|---|---|---|---|
| | Upperbound [14] | – | – | 77.7 |
| | Baseline* [7] | 47.3 | 62.7 | 61.4 |
| (A1) | $M_t \leftarrow$ Seg. Mask | 47.0 | 61.9 | 61.1 |
| (A2) | Round = 1 | 47.7 | 63.5 | 62.2 |
| | Round = 2 | **48.0** | **64.0** | **62.8** |
| | Round = 3 | 47.8 | 63.8 | 62.5 |
| | Round = 4 | 47.4 | 63.5 | 62.1 |
| (A3) | Block-2 | 47.4 | 62.9 | 61.7 |
| | Block-3 | 47.6 | 63.2 | 62.1 |
| | Block-4 | 47.9 | 63.7 | 62.6 |
| | Block-5 | **48.0** | **64.0** | **62.8** |
| | Dense | 47.8 | 63.8 | 62.7 |
| (A4) | $C_{\text{Pseudo-Mask}}$ | 47.8 | 63.6 | 62.5 |
| | $C_{\text{Seg. Mask}}$ | **48.0** | **64.0** | **62.8** |

Table A2: Ablations of DSRG [7]+CONTA on PASCAL VOC 2012 [5] in mIoU (%). "*" denotes our re-implemented results. "Seg. Mask" refers to the segmentation mask of the *val* set. "–" denotes that the result is N.A. for the fully-supervised model.

| | Setting | CAM | Pseudo-Mask | Seg. Mask |
|---|---|---|---|---|
| | Upperbound [14] | – | – | 71.6 |
| | Baseline* [10] | 46.5 | 53.4 | 50.7 |
| (A1) | $M_t \leftarrow$ Seg. Mask | 46.4 | 53.1 | 50.3 |
| (A2) | Round = 1 | 47.1 | 54.3 | 51.7 |
| | Round = 2 | 47.6 | 55.1 | 52.6 |
| | Round = 3 | **47.9** | **55.7** | **53.2** |
| | Round = 4 | 47.7 | 55.6 | 53.0 |
| (A3) | Block-2 | 46.8 | 53.9 | 51.2 |
| | Block-3 | 47.1 | 54.5 | 51.5 |
| | Block-4 | 47.6 | 55.1 | 52.4 |
| | Block-5 | **47.9** | **55.7** | **53.2** |
| | Dense | 47.8 | 55.6 | 53.0 |
| (A4) | $C_{\text{Pseudo-Mask}}$ | 47.7 | 55.3 | 52.9 |
| | $C_{\text{Seg. Mask}}$ | **47.9** | **55.7** | **53.2** |

Table A3: Ablations of SEC [10]+CONTA on PASCAL VOC 2012 [5] in mIoU (%). "*" denotes our re-implemented results. "Seg. Mask" refers to the segmentation mask of the *val* set. "–" denotes that the result is N.A. for the fully-supervised model.

## 5 More Visualizations

This is Appendix to Section 4.4 "Comparison with State-of-the-arts". More segmentation results are visualized in Figure A1. We can observe that most of our resulting masks are of high quality. The segmentation masks predicted by SEAM+CONTA are more accurate and have better integrity, *e.g.*, for cow, horse, bird, person lying next to the dog, and person standing next to the cows. In particular, SEAM+CONTA works better to prediction the edges of some thin objects or object parts, *e.g.*, the tail (or the head) of bird, car, and person in the car.

| | Setting | CAM | Pseudo-Mask | Seg. Mask |
|---|---|---|---|---|
| | Upperbound* [14] | – | – | 44.8 |
| | Baseline* [19] | 25.1 | 31.5 | 31.9 |
| (A1) | $M_t \leftarrow$ Seg. Mask | 24.8 | 31.1 | 31.4 |
| (A2) | Round = 1 | 25.7 | 31.9 | 32.4 |
| | Round = 2 | 26.2 | 32.2 | 32.7 |
| | Round = 3 | **26.5** | **32.5** | **32.8** |
| | Round = 4 | 26.3 | 32.1 | 32.6 |
| (A3) | Block-2 | 25.7 | 32.0 | 32.3 |
| | Block-3 | 25.9 | 32.1 | 32.4 |
| | Block-4 | 26.3 | 32.4 | 32.6 |
| | Block-5 | **26.5** | **32.5** | **32.8** |
| | Dense | **26.5** | 32.4 | 32.5 |
| (A4) | $C_{\text{Pseudo-Mask}}$ | 26.4 | 32.0 | 32.6 |
| | $C_{\text{Seg. Mask}}$ | **26.5** | **32.5** | **32.8** |

Table A4: Ablation results of SEAM [19]+CONTA on MS-COCO [13] in mIoU (%). "*" denotes our re-implemented results. "Seg. Mask" refers to the segmentation mask of the *val* set. "–" denotes that the result is N.A. for the fully-supervised model.

| | Setting | CAM | Pseudo-Mask | Seg. Mask |
|---|---|---|---|---|
| | Upperbound* [14] | – | – | 42.5 |
| | Baseline* [1] | 27.4 | 34.0 | 32.6 |
| (A1) | $M_t \leftarrow$ Seg. Mask | 27.1 | 33.5 | 32.3 |
| (A2) | Round = 1 | 28.0 | 34.3 | 32.9 |
| | Round = 2 | 28.4 | 34.8 | 33.2 |
| | Round = 3 | **28.7** | **35.2** | **33.4** |
| | Round = 4 | 28.5 | 35.0 | 33.2 |
| (A3) | Block-2 | 27.7 | 34.3 | 32.8 |
| | Block-3 | 27.9 | 34.5 | 32.9 |
| | Block-4 | 28.4 | 34.9 | 33.2 |
| | Block-5 | **28.7** | **35.2** | **33.4** |
| | Dense | 28.6 | **35.2** | 33.1 |
| (A4) | $C_{\text{Pseudo-Mask}}$ | 28.5 | 35.0 | 33.2 |
| | $C_{\text{Seg. Mask}}$ | **28.7** | **35.2** | **33.4** |

Table A5: Ablation results of IRNet [1]+CONTA on MS-COCO [13] in mIoU (%). "*" denotes our re-implemented results. "Seg. Mask" refers to the segmentation mask of the *val* set. "–" denotes that the result is N.A. for the fully-supervised model.

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

| | Setting | CAM | Pseudo-Mask | Seg. Mask |
|---|---|---|---|---|
| | Upperbound [14] | – | – | 45.0 |
| | Baseline* [7] | 19.8 | 26.1 | 25.6 |
| **(A1)** | $M_t \leftarrow$ Seg. Mask | 19.5 | 25.9 | 25.5 |
| **(A2)** | Round = 1 | 20.5 | 26.9 | 26.1 |
| | Round = 2 | **20.9** | **27.5** | **26.4** |
| | Round = 3 | 20.7 | 27.2 | 26.2 |
| | Round = 4 | 20.4 | 26.9 | 26.0 |
| **(A3)** | Block-2 | 20.1 | 26.8 | 25.9 |
| | Block-3 | 20.2 | 27.0 | 26.0 |
| | Block-4 | 20.5 | 27.2 | 26.2 |
| | Block-5 | **20.9** | **27.5** | **26.4** |
| | Dense | 20.8 | 27.3 | 26.1 |
| **(A4)** | $C_{\text{Pseudo-Mask}}$ | 20.7 | 27.2 | 26.1 |
| | $C_{\text{Seg. Mask}}$ | **20.9** | **27.5** | **26.4** |

Table A6: Ablation results of DSRG [7]+CONTA on MS-COCO [13] in mIoU (%). "*" denotes our re-implemented results. "Seg. Mask" refers to the segmentation mask of the *val* set. "–" denotes that the result is N.A. for the fully-supervised model.

| | Setting | CAM | Pseudo-Mask | Seg. Mask |
|---|---|---|---|---|
| | Upperbound [14] | – | – | 41.0 |
| | Baseline* [10] | 18.7 | 24.0 | 22.4 |
| **(A1)** | $M_t \leftarrow$ Seg. Mask | 18.1 | 23.5 | 21.2 |
| **(A2)** | Round = 1 | 20.1 | 24.4 | 23.0 |
| | Round = 2 | 21.2 | 24.7 | 23.4 |
| | Round = 3 | **21.8** | **24.9** | **23.7** |
| | Round = 4 | 21.4 | 24.5 | 23.5 |
| **(A3)** | Block-2 | 19.5 | 24.2 | 22.7 |
| | Block-3 | 19.9 | 24.4 | 22.9 |
| | Block-4 | 20.6 | 24.7 | 23.5 |
| | Block-5 | **21.8** | **24.9** | **23.7** |
| | Dense | **21.8** | 24.6 | 23.5 |
| **(A4)** | $C_{\text{Pseudo-Mask}}$ | 21.5 | 24.7 | 23.4 |
| | $C_{\text{Seg. Mask}}$ | **21.8** | **24.9** | **23.7** |

Table A7: Ablation results of SEC [10]+CONTA on MS-COCO [13] in mIoU (%). "*" denotes our re-implemented results. "Seg. Mask" refers to the segmentation mask of the *val* set. "–" denotes that the result is N.A. for the fully-supervised model.

[3] Pierre Baldi and Peter Sadowski. The dropout learning algorithm. *Artificial Intelligence*, 210:78–122, 2014.

[4] Jia Deng, Wei Dong, Richard Socher, Li-Jia Li, Kai Li, and Li Fei-Fei. Imagenet: A large-scale hierarchical image database. In *CVPR*, 2009.

[5] Mark Everingham, SM Ali Eslami, Luc Van Gool, Christopher KI Williams, John Winn, and Andrew Zisserman. The pascal visual object classes challenge: A retrospective. *IJCV*, 111(1):98–136, 2015.

[6] Kaiming He, Xiangyu Zhang, Shaoqing Ren, and Jian Sun. Deep residual learning for image recognition. In *CVPR*, 2016.

[7] Zilong Huang, Xinggang Wang, Jiasi Wang, Wenyu Liu, and Jingdong Wang. Weakly-supervised semantic segmentation network with deep seeded region growing. In *CVPR*, 2018.

[8] Huaizu Jiang, Jingdong Wang, Zejian Yuan, Yang Wu, Nanning Zheng, and Shipeng Li. Salient object detection: A discriminative regional feature integration approach. In *CVPR*, 2013.

[9] Diederik P Kingma and Jimmy Ba. Adam: A method for stochastic optimization. In *ICLR*, 2015.

[10] Alexander Kolesnikov and Christoph H Lampert. Seed, expand and constrain: Three principles for weakly-supervised image segmentation. In *ECCV*, 2016.

[11] Philipp Krähenbühl and Vladlen Koltun. Efficient inference in fully connected crfs with gaussian edge potentials. In *NeurIPS*, 2011.

[12] Alex Krizhevsky, Ilya Sutskever, and Geoffrey E Hinton. Imagenet classification with deep convolutional neural networks. In *NeurIPS*, 2012.

[13] Tsung-Yi Lin, Michael Maire, Serge Belongie, James Hays, Pietro Perona, Deva Ramanan, Piotr Dollár, and C Lawrence Zitnick. Microsoft coco: Common objects in context. In *ECCV*, 2014.

[14] Jonathan Long, Evan Shelhamer, and Trevor Darrell. Fully convolutional networks for semantic segmentation. In *CVPR*, 2015.

[15] Judea Pearl. *Causality: Models, Reasoning and Inference*. Springer, 2000.

[16] Judea Pearl, Madelyn Glymour, and Nicholas P Jewell. *Causal inference in statistics: A primer*. John Wiley & Sons, 2016.

[17] Abhinav Shrivastava, Abhinav Gupta, and Ross Girshick. Training region-based object detectors with online hard example mining. In *CVPR*, 2016.

[18] Karen Simonyan and Andrew Zisserman. Very deep convolutional networks for large-scale image recognition. In *ICLR*, 2015.

[19] Yude Wang, Jie Zhang, Meina Kan, Shiguang Shan, and Xilin Chen. Self-supervised equivariant attention mechanism for weakly supervised semantic segmentation. In *CVPR*, 2020.

[20] Zifeng Wu, Chunhua Shen, and Anton Van Den Hengel. Wider or deeper: Revisiting the resnet model for visual recognition. *Pattern Recognition*, 90(1):119–133, 2019.

[21] Kelvin Xu, Jimmy Ba, Ryan Kiros, Kyunghyun Cho, Aaron Courville, Ruslan Salakhudinov, Rich Zemel, and Yoshua Bengio. Show, attend and tell: Neural image caption generation with visual attention. In *ICML*, 2015.

[22] Fisher Yu and Vladlen Koltun. Multi-scale context aggregation by dilated convolutions. In *ICLR*, 2016.

[23] Bolei Zhou, Aditya Khosla, Agata Lapedriza, Aude Oliva, and Antonio Torralba. Learning deep features for discriminative localization. In *CVPR*, 2016.

|   |   |   |   |
|:-:|:-:|:-:|:-:|
| **Image** | **SEAM** | **SEAM+CONTA** | **Ground-Truth** |

Figure A1: More visualization results. Samples are from PASCAL VOC 2012 [5]. Red rectangles highlight the improved regions predicted by SEAM [19]+CONTA.