[Reviews · NeurIPS 2020]

Review 1

Summary and Contributions: The paper proposes an interesting method for weakly supervised semantic segmentation based on a casual intervention model. The context prior plays an important role in semantic segmentation. The model uses the context prior derived from the object mask distribution predicted on the whole dataset to help learn better pseudo semantic labels. The experiments are conducted on both Pascal VOC 2012 and COCO dataset, showing the effectiveness of the proposed approach.

Strengths: The pros: - The paper is clearly motivated and very well written, and is easy to follow as well. - The idea of using context prior for iteratively improving learning the semantic pseudo labels are interesting and novel. The ablation study shows that the proposed scheme could effectively boost the performance over two very strong baseline models with clear gain. - The final results of the model shows the state-of-the-art performance on both Pascal-VOC and COCO datasets.

Weaknesses: The cons: - The calculation of the context prior is not very clearly described. How is it derived from the whole prediction distribution of the classification model? - The effectiveness of using the context prior is not very demonstrated to the reviewer. How about disabling the context prior and just using the prediction masks to directly contact with the input image X? This seems to be an important baseline to run. - The context prior is actually help to improve the prediction of the pseudo lables. Another interesting baseline is to refine the predicted semantic label map with a CRF model, and then concat the refined one with input image X. This is to show the context prior is indeed more beneficial than another simple way to provide semantic prior.

Correctness: correct

Clarity: yes

Relation to Prior Work: yes

Reproducibility: Yes

Additional Feedback: Please refer to the weakness part for the rebuttal.


Review 2

Summary and Contributions: The paper addresses the problem of explicitly modelling spatial context in weakly-supervised semantic segmentation (supervision by image-level labels). While most modern methods rely on implicit learning of context, that may result in learning spurious correlations inherent to a specific dataset, e.g. that horse can only be present when a person is also present. The paper uses Pearl’s causal inference framework to unbias training data. Experiments explore different ways to model the context and evaluate design choices, in addition to improving SOTA in weakly-supervised segmentation on PASCAL VOC and MS-COCO.

Strengths: Novelty. The paper is the first one to apply causal inference to semantic segmentation in order to explicitly leverage the context. Motivation. While, as the paper mentions, all modern methods have to take advantage of the context, the correlations in the training data may be not representative for the deployment scenario, so a causal model is reasonable to use (I think the paper undersells this contribution; see below for suggestions to test better generalisation). Causal inference is even more helpful in the weakly-supervised case, where the model cannot even learn spatial correlations implicitly. From the vision side, the paper is motivated by Marr’s work on visual psychology and the further work on modelling context like DPMs. Relevance. The paper is relevant to a large part of NeurIPS community, i.e. both to causal-inference and computer-vision people. Experiments. They are convincing. The paper improves SOTA on two datasets, tested different versions of the system on them, and used different segmentation backbones. The backdoor adjustment consistently improves the results by 1–2 p.ps. Specifically, the Q1 check (line 254) is interesting; it proves that the improvement is not just due to better segmentation result. The paper is generally clear and seems correct (see below for more details).

Weaknesses: Not really a weakness, but something that can make the paper even stronger. Since the paper claims to remove spurious correlation inherent to the dataset, it would be nice to test if the model generalises better in the transfer-learning scenario: train on one dataset, and predict on another. See the following sections for the comments on correctness and clarity. UPD. The rebuttal addressed my main concerns. Re Q2, by no means I suggested to use Rubin’s framework; it is the one I am more familiar with. What I think is that it will be a valuable addition to the paper if it states the assumptions of the causal framework of choice, and explains what it means in context of the task. I imagine the paper will influence future computer-vision research attempting to use causal inference; it will be important for the followers to understand the assumptions.

Correctness: The paper does not clearly state the assumptions related to applying the backdoor adjustment. I am more familiar with Rubin’s framework, which has the overlap assumption; I think something similar should be required for the backdoor: in order to re-weight horses without riders, do you need to have at least one example of a horse without person in the training data? Stating the assumptions explicitly would be nice. In line 199, the paper refers to Appendix for the assumptions. There is derivation but the assumptions are not explicitly stated there. For the derivation, is it the example for 2 classes and not the general case? Figure 6 shows examples with better masks but I don’t see how they illustrate the hypothesis. It would be good to inspect the cases where the objects occur in an unusual context. E.g. a cow on grass is probably not the case when the proposed methods shines.

Clarity: The paper is generally clear and easy to read. I can’t follow the claim in the lines 141–144 that no context can contribute to Y when training P(Y | X). The dataset biases are present in X, so the model can still learn the context from them? Please clarify this claim. The description of Step 2 – Pseudo-Mask Generation feels vague; more formal description would help. In Table 1, it is not clear what pseudo-mask column refers to. Is it the inferred mask on the last round? Should it converge to the Segmentation mask and have the same accuracy on the last rounds?

Relation to Prior Work: Related work is comprehensive.

Reproducibility: Yes

Additional Feedback: Typos: line 36: As you might concern ← you might be concerned (also this sounds a bit manipulatory); line 65: random control trial ← randomised control trial; line 68: in an unified ← in a unified; line 71: the prohibitively “physical” intervention ← the prohibitively expensive “physical” intervention?; line 195–: the use of X_m for segmentation masks may be confusing, as X is used for input images. In experiments, please use percentage points (p.p.) to describe the change in percentage, to avoid the ambiguity.


Review 3

Summary and Contributions: 1. This paper analyzes three problems in pseudo-mask generation in weakly supervised semantic segmentation, including object ambiguity, incomplete background, incomplete foreground, and points out that these problems are due to the context prior in dataset. 2. This paper proposes the context adjustment to remove the confounding effect of context prior and adopts an iterative procedure to generate high-quality seed areas.

Strengths: 1. This paper analyzes the phenomenon of context bias in WSSS and the problems it causes in detail, which is somewhat novel. 2. The proposed method achieves great performance improvement and works well over different methods and datasets.

Weaknesses: 1. If M_{t} is calculated only using $C$ without $X$, how about the results? This experiment should be added to demonstrate the necessity of $X$. 2. The description of Equation 3 is not very clear. X_m \in R^{hw x n} and C \in R^{hw x n}, so the part contained in softmax function has the shape of nxn. Then which dimension is the softmax function applied for? Why use sqrt(n)? If all P(c)=1/n, the \sigma_c and P(c) can be removed, but the output M_{t+1} is no longer a single channel map. Is W_{1} and W_{2} learned or manually set? Therefore Equation 3 should be rewritten carefully or it would be confused. ----------------------- Reviewer has read the author response. I would like to stick to my original score.

Correctness: Yes

Clarity: Yes

Relation to Prior Work: Yes

Reproducibility: Yes

Additional Feedback: Please refer to the weakness


Review 4

Summary and Contributions: This paper tackles the challenging weakly-supervised semantic segmentation task. More specifically, the authors establish a structural causal model to handle the negative impact caused by the scene context. The proposed method is model-agnostic, showing consistent improvement on top of several methods on VOC and COCO datasets.

Strengths: 1. Experiments 1) The proposed method establishes a new state-of-the-art on both VOC and COCO datasets 2) The proposed method serves as a plug-and-play module, consistently improving several strong baselines in a model-agnostic way 3) By detailed ablation study, the authors well validate their design choices 2. Novelty Although the causal intervention method (backdoor adjust) is not new, this paper is the first to introduce causal inference into the weakly-supervised semantic segmentation task. I think this paper is novel. 3. Relevance Causal inference is definitely relevant to the NeurIPS community. And this paper well demonstrates how to apply causal inference to tackle a challenging computer vision problem.

Weaknesses: 1. Questions about the structural causal model 1) I feel that the confounder set C can be interpreted as “object shapes and where to place them”. But I still do not have an intuitive way to interpret the image-specific context representation M. 2) Why is X -> M instead of M -> X? From my understanding, we sample object shapes and their locations to get M. And then later we sample object appearance (e.g., texture, lighting, etc.) to get X. 2. Implementation 1) Since the images in both VOC and COCO have different sizes and ratios, I wonder how the authors construct the confounder set C. 2) Is the segmentation mask X_m (L195) logits or probabilities? 3) I feel a bit confused about Eqn. (3). It seems that W_1 and W_2 are used as projection matrices, reducing the dimension from original spatial size (hw) to the number of class (n). I wonder if this is reasonable. And I think the projected embedding space can be any dimension, not necessarily to be n? 4) Why do the authors choose P(c) to be uniform? Using the actual object frequencies in the dataset to represent P(c) might be better? 3. Experiments 1) For Q1 in Table 1, more details are required. How exactly the segmentation mask is used in the network? What’s the dimension? Is it a soft mask with probability/logit, or a binary mask with one-hot label? What if the author constructed a self-attention mask similar to Eqn. (3)? 2) Since the proposed method requires iterative refinement, I think it should also compare with the Noisy Student training [A1]. For example, after the first time training of the segmentation model, the authors can then use it to generate pseudo labels. And then, use the pseudo labels to re-train the segmentation model. By comparing with this baseline, we can then know if the performance gain comes from causal intervention or simply from the iterative refinement of the segmentation model itself. 3) In Table, it seems that the proposed method has smaller gain when using stronger feature backbone. Does it mean that, stronger network can better handle the context (e.g., effectively exploit its advantage while discard its negative impact)? Reference [A1] Xie et al. CVPR 2020. Self-training with Noisy Student improves ImageNet classification

Correctness: My only concern is the causal link direction between M and X in the structural causal model. Please check the weakness section.

Clarity: Yes. Overall, I find this paper easy to read and understand.

Relation to Prior Work: Yes. The proposed method can be regarded as a plug-and-play module to be incorporated with existing state-of-the-art weakly-supervised semantic segmentation methods.

Reproducibility: Yes

Additional Feedback: Please address the raised issues in the weakness section. Update: The rebuttal addresses most of the raised issues. Thus, I am willing to increase my rating and recommend acceptance.

[Author Response · NeurIPS 2020]

We thank all the anonymous reviewers for their constructive feedback. We address each comment as follows.

**R1-Q1:The calculation of context prior.** Sorry for the confusion. In our work, the context prior is defined as a
confounder set $C = \{c_1, c_2, ..., c_n\}$, where $n$ is the class size in dataset. Each $c$ is the $h \times w$ average segmentation
mask of the $i$-th class images, which is obtained from the trained segmentation model in the last round (line 159-162).

**R1-Q2:Just using the predicted mask to concat.** Sorry for the unclear presentation. By (**Q1**) in Table 1 of the main
paper, we directly concat the predicted mask (*i.e.*, Seg.Mask in Table 1) into the backbone network. Experimental
results show that just using the predicted mask without $C$ is even worse than the baseline SEAM [46].

**R1-Q3:Refine the predicted mask with CRF.** We followed your suggestion to use the CRF to refine the predicted
mask, then concat the refined mask into the backbone network for a new round classification. Results on the baseline
SEAM show that CRF (*vs* CONTA) is only effective in the first round, *i.e.*, achieved at most 0.2 (*vs* 1.1)%, 0.3 (*vs* 2.3)%
and 0.3 (*vs* 1.8)% mIoU improvements for CAM, pseudo-mask and segmentation mask, respectively.

**R2-Q1:Test in the transfer-learning scenario.** We followed your suggestion to train models on COCO and test on
PASCAL. Results on the *val* set show that the baseline SEAM (*vs* +CONTA) can achieve 32.1 (*vs* 33.2)% mIoU.

**R2-Q2:The assumption of backdoor adjustment.** Its identifiability assumes that the confounder set is fully ob-
served, *e.g.*, a ground-truth vocabulary of contexts in our visual world. Unfortunately, it is impossible in prac-
tice and thus CONTA requires an iterative "guess" of the hidden confounder. Therefore, at each iteration, we
need what you suggested: "one example of horse (person) without person (horse)", or more generally, "one ex-
ample of class A without B", to disentangle A and B. Fortunately, it is feasible in the PASCAL and COCO
datasets. We will follow your suggestion to revisit CONTA in the Rubin's potential outcome framework in revision.

**R2-Q3:Unusual context.** In fact, usual context such as
"cow on grass" is better to illustrate, as the object co-
occurrence is more confounding in WSSS (line 43-49).
We also show an unusual example of "car crashed in water"
in Figure R1. We will highlight this in revision.

**Input**     **SEAM**     **SEAM+CONTA**     **Ground-Truth**

Figure R1: An unusual example: car crashed in water.

**R2-Q4:The assumption and derivation in Appendix 2.**
We will include the assumption of NWGM approximation for self-contained purpose in revision, as in VC R-CNN
[45]. For the derivation, we only derived $s_1(\cdot)$: the positive class term. Besides, we can also obtain derivation for the
negative term $s_2(\cdot)$ through the similar process. We will clarify it in revision.

**R2-Q5:No context can contribute to $Y$ when training $P(Y|X)$?** Sorry for the confusion. We mean that if the path
$M \to Y$ is **NOT** existing, then no context can contribute to $Y$, and we can never recover the seed areas in WSSS by
training $P(Y|X)$. But the reality is that we can recover the seed areas, which indicates that $M$ is existing.

**R2-Q6** and **R4-Q1:Typos/visualizations/vague statements/suggestions.** We are grateful for your constructive sug-
gestions. We will revise our paper including typos, visualizations, and vague statements according to your suggestions.

**R3-Q1:Only using $C$ without $X_m$.** We followed your suggestion to directly concat $C$ corresponding to the image
class into the backbone network. Experimental results show that only using $C$ (*vs* the baseline SEAM) achieves 54.6
(*vs* 55.1)%, 62.1 (*vs* 63.1)% and 63.6 (*vs* 64.3)% mIoU on CAM, pseudo-mask and segmentation mask, respectively.

**R3-Q2** and **R4-Q4:The description of Eq.3.** Sorry for the confusion. The segmentation mask $X_m \in \mathbb{R}^{hw \times n}$ denotes
the logits. In our implementation of Eq.(3), $X_m$ is first reshaped into a $hw \times 1$ vector. Therefore, the part contained in
the softmax function has the shape of $1 \times n$. Following [45], $\sqrt{n}$ is used as a constant scaling factor for normalization.
$\mathbf{W}_1$ and $\mathbf{W}_2$ are two learnable projection matrices. We will revise the writing of Eq.(3) in revision.

**R3-Q3** and **R4-Q6:About P(c).** We are sorry for a typo here. $\sum_c$ and $P(c)$ can not be removed in Eq.(3), because
each $P(c)$ corresponds to a specific entry in the confounder set $C(c)$. We will fix this typo in revision. Besides, the
reason why we choose $P(c)$ to be uniform $1/n$ is that CONTA is designed to go beyond the dataset. If we use the
actual class frequencies to represent $P(c)$, CONTA will be still confounded by the dataset observation.

**R4-Q2:$X \to M$ or $M \to X$?** In our assumption, $M$ is the image-specific representation ($X \to M$), in the form of
linear combination of context masks (Eq.3), which is certainly regularized by the context ($C \to M$). Therefore, what
you think of "sampling object shapes and location" and "sampling object appearance" actually correspond to $C \to M$
and $C \to X$ in our model.

**R4-Q3:Construct the confounder set.** Sorry for the confusion. In our implementation, the input image is first resized
into a fixed scale before feeding into the network. For example, we set $448 \times 448$ for SEAM+CONTA, and $512 \times 512$
for IRNet [1]+CONTA. Therefore, each of the entry in the confounder set follows the same scale as the input image.

**R4-Q5:The projected embedding space can be any dimension?** No, the projected embedding space can not be set
to other dimensions, because $n$ in $\mathbf{W}_1$ and $\mathbf{W}_2$ corresponds to the class size in dataset, which has the same size as the
confounder set $C$. We have no reason to set $n$ to other dimensions.

**R4-Q7:Use pseudo-labels to re-train the segmentation model.** We followed your suggestion to use pseudo-labels to
re-train the segmentation model. Experimental results show that using pseudo-labels (*vs* the baseline SEAM) achieves
62.7 (*vs* 64.3)% mIoU. **R4-Q8:Stronger backbone can better handle context?** First, different backbones correspond
to different seed generation or expansion methods. Therefore, we can not draw this conclusion. Second, this conjecture
may be correct. Because the stronger backbone indeed locates object areas more accurately than a weaker one.

[Meta-Review · NeurIPS 2020]

This paper proposes using a causal inference framework for weakly supervised semantic segmentation. It corrects mistakes in pseudomasks by adjusting for confounding effects. By relying on causal inference, as opposed to a discriminative model, the goal is to avoid relying on spurious correlations in the training data that might fail to generalize. The reviewers agree that using backdoor adjustments for semantic segmentation is a novel use of the technique, and that the experimental results are impressive. One suggestion for improvement in the camera ready is to more clearly state the modeling assumptions of the causal framework that is used, and to elaborate on what their implications are for this problem.